# Long-Term Protective Effects of Succinate Dehydrogenase Inhibition during Reperfusion with Malonate on Post-Infarction Left Ventricular Scar and Remodeling in Mice

**DOI:** 10.3390/ijms25084366

**Published:** 2024-04-15

**Authors:** Laura Valls-Lacalle, Marta Consegal, Freddy G. Ganse, Laia Yáñez-Bisbe, Javier Pastor, Marisol Ruiz-Meana, Javier Inserte, Begoña Benito, Ignacio Ferreira-González, Antonio Rodríguez-Sinovas

**Affiliations:** 1Cardiovascular Diseases Research Group, Vall d’Hebron University Hospital and Research Institute, 08035 Barcelona, Spain; lvalls92@gmail.com (L.V.-L.); martaconse@gmail.com (M.C.); freddy.ganse@vhir.org (F.G.G.); laia.yanez@vhir.org (L.Y.-B.); javpasbau@gmail.com (J.P.); marisol.ruizmeana@vhir.org (M.R.-M.); javier.inserte@vhir.org (J.I.); begona.benito@vhir.org (B.B.); iferregon@gmail.com (I.F.-G.); 2Centro de Investigación Biomédica en Red de Enfermedades Cardiovasculares (CIBERCV), Instituto de Salud Carlos III, 28029 Madrid, Spain; 3Cardiology Department, Hospital Universitari Vall d’Hebron, 08035 Barcelona, Spain; 4Department of Medicine, Universitat Autònoma de Barcelona, 08035 Barcelona, Spain; 5Centro de Investigación Biomédica en Red en Epidemiología y Salud Pública (CIBERESP), Instituto de Salud Carlos III, 28029 Madrid, Spain

**Keywords:** succinate dehydrogenase, ischemia–reperfusion, malonate, remodeling, scar, mitochondria

## Abstract

Succinate dehydrogenase inhibition with malonate during initial reperfusion reduces myocardial infarct size in both isolated mouse hearts subjected to global ischemia and in in situ pig hearts subjected to transient coronary ligature. However, the long-term effects of acute malonate treatment are unknown. Here, we investigated whether the protective effects of succinate dehydrogenase inhibition extend to a reduction in scar size and adverse left ventricular remodeling 28 days after myocardial infarction. Initially, ten wild-type mice were subjected to 45 min of left anterior descending coronary artery (LAD) occlusion, followed by 24 h of reperfusion, and were infused during the first 15 min of reperfusion with saline with or without disodium malonate (10 mg/kg/min, 120 μL/kg/min). Malonate-treated mice depicted a significant reduction in infarct size (15.47 ± 3.40% of area at risk vs. 29.34 ± 4.44% in control animals, *p* < 0.05), assessed using triphenyltetrazolium chloride. Additional animals were then subjected to a 45 min LAD ligature, followed by 28 days of reperfusion. Treatment with a single dose of malonate during the first 15 min of reperfusion induced a significant reduction in scar area, measured using Picrosirius Red staining (11.94 ± 1.70% of left ventricular area (n = 5) vs. 23.25 ± 2.67% (n = 9), *p* < 0.05), an effect associated with improved ejection fraction 28 days after infarction, as determined using echocardiography, and an attenuated enhancement in expression of the pro-inflammatory and fibrotic markers NF-κB and Smad2/3 in remote myocardium. In conclusion, a reversible inhibition of succinate dehydrogenase with a single dose of malonate at the onset of reperfusion has long-term protective effects in mice subjected to transient coronary occlusion.

## 1. Introduction

Restoration of coronary blood flow following transient coronary occlusion is accompanied by a burst of reactive oxygen species (ROS) [1], which play a key role in reperfusion injury. ROS contribute to the opening of the mitochondrial permeability transition pore (MPTP) during reperfusion and to cardiomyocyte hypercontrature, the two primary mechanisms responsible for cell death during myocardial infarction [2,3]. The main sources of ROS in postischemic tissues include xanthine oxidase, NADPH oxidase, nitric oxide synthase, and notably, the mitochondria [4]. Under physiological conditions, electron transport to oxygen, occurring in this organelle, is closely coupled with oxidative phosphorylation to generate ATP. However, the low oxygen tension present in a reperfused myocardium causes some electrons to leak from mitochondrial complexes I and III, resulting in the generation of superoxide [1,5]. The involvement of ROS in reperfusion injury has been confirmed by studies using antioxidants or ROS scavengers, which, when given at the onset of reperfusion, have been able to reduce infarct size in various experimental models [4,6]. However, clinical trials testing the efficacy of antioxidants in patients with cardiovascular disease have not yielded conclusive results [7,8]. A reduction in oxidative stress can also be achieved through direct inhibition of mitochondrial respiratory complexes, as has been demonstrated with drugs targeting complex I [9,10]. Moreover, recent studies have suggested that, in addition to complexes I and III, mitochondrial complex II may also play a significant role in ROS production after ischemia [11,12]. Supporting this notion, it has been demonstrated that myocardial levels of succinate, the endogenous substrate of complex II (or succinate dehydrogenase, an enzyme also involved in the tricarboxylic acid cycle), are markedly enhanced during ischemia, and that treatments aimed at preventing this increase can reduce infarct size [13,14]. The mechanisms involved in this protective effect include the rapid oxidation of accumulated succinate during initial reperfusion, leading to a reverse electron transfer from mitochondrial complex II to complex I and ROS production by the latter [13,14].

However, application of adjunctive treatments before ischemia is not feasible in patients with ST segment elevation myocardial infarction (STEMI). In this regard, we have recently demonstrated that reversible and transient inhibition of mitochondrial complex II with disodium malonate, given at the onset of reperfusion, is able to reduce infarct size in both isolated mouse hearts [15] and in pigs subjected to transient coronary occlusion, with the treatment being selectively applied to the area at risk [16]. This protective effect was associated with reduced succinate oxidation during reperfusion, decreased oxidative stress, and attenuated MPTP opening [15,16].

Unlike the scenario involving a transient coronary occlusion followed by reperfusion, others have failed to demonstrate a reduction in infarct size when malonate was given daily for three days and began within the first hour after ischemia onset in mice subjected to permanent coronary ligature (thus lacking reperfusion) [17]. In contrast, under these conditions, chronic treatment with malonate for 14 days was able to promote adult cardiomyocyte proliferation and cardiac regeneration 4 weeks after coronary occlusion [17].

In this context, it remains unclear whether the protective effect of malonate, when administered during reperfusion as a single dose, persists long after ischemia and whether it may influence post-infarction left ventricular remodeling. Therefore, the aim of this work has been to analyze the effects of reversible inhibition of mitochondrial complex II with a single dose of disodium malonate on post-infarction scar size and ventricular remodeling in an in vivo mouse model of transient coronary occlusion (45 min) followed by 28 days of reperfusion.

## 2. Results

There were no deaths between animals subjected to 45 min of LAD occlusion and 24 h of reperfusion (n = 5/group). In contrast, among the 21 mice reperfused for 28 days, 4 died during follow-up (3 in the control group and 1 in the malonate-treated group). In addition, one animal from the malonate-treated group was excluded due to a lack of reperfusion. Finally, two more animals from this group were excluded due to poor LAD visualization during surgery, with repeated attempts at occluding the artery resulting in extensive myocardial injury. Thus, the final number of individuals included in the study was nine for the control group and five for the malonate-treated group.

### 2.1. Short-Term Effects of Reversible Succinate Dehydrogenase Inhibition with Malonate during Reperfusion on Myocardial Infarct Size

Malonate administration at the onset of reperfusion induced a reduction in myocardial infarct size, measured using 2,3,5-triphenyltetrazolium chloride staining 24 h after reperfusion in mice subjected to transient coronary occlusion (Figure 1A). No differences were observed in the size of the area at risk (Figure 1B).

### 2.2. Long-Term Effects of Reversible Succinate Dehydrogenase Inhibition with Malonate during Reperfusion on Scar Size and Post-Infarction Left Ventricular Remodeling

Treatment with a single dose of malonate during the first 15 min of reperfusion induced a significant reduction in scar area, measured using Picrosirius Red staining 28 days after LAD occlusion (Figure 2A,B). Similarly, interstitial collagen deposition in the remote myocardium (Figure 2A,C) and perivascular fibrosis (Figure 2A,D) were also diminished by malonate treatment. These protective effects were associated with a significantly attenuated deterioration in myocardial function (i.e., EF and FS) in malonate-treated animals (two-way ANOVA, *p* = 0.005 and 0.012 for interaction between time and treatment for EF and FS, respectively), as compared with that occurring in control mice treated with saline (Figure 3 and Figure 4). Repeated measures ANOVA demonstrated a significantly lower body weight in treated animals and a significant effect of time, but there was no interaction between both variables, suggesting that body weight was similarly reduced during the 28-day experimental period in both groups (Figure 2E).

### 2.3. Expression of Pro-Inflammatory and Fibrotic Markers in the Remote Myocardium

Western blot analysis demonstrated significantly enhanced expression in the remote myocardium of control animals subjected to LAD occlusion followed by 28 days after reperfusion of NF-κB and Smad2/3, together with increased levels of phosphorylated Smad2/3, as compared with sham-operated animals. The enhancement in NF-κB and Smad2/3 was attenuated in animals treated with malonate (Figure 5).

Expression of α-smooth muscle actin was highly variable in the remote myocardium of both sham-operated mice and in animals subjected to transient coronary occlusion and 28 days of reperfusion. Despite this fact, a significant reduction in α-smooth muscle actin expression was observed in mice subjected to LAD occlusion, independently of group allocation, as compared with sham-operated animals (Figure 6). No differences in fibroblast activation protein α (FAPα) were observed between groups (Figure 6).

RT-PCR analysis demonstrated a significant increase in mRNA levels for COL1A1 in myocardial samples from infarcted animals as compared with sham-operated mice. Similar trends were observed for other proteins involved in collagen synthesis (COL3A1, TGFβ1, and ACTA2). However, contrary to Western blot data, no differences were observed between samples from infarcted animals treated with or without malonate (Figure 7). Furthermore, no significant differences were observed in mRNA levels for IL-1β, whereas a significant increase in TNFα was detected in those animals receiving malonate as compared with those receiving saline (Figure 7).

## 3. Discussion

This study demonstrates that a transient and reversible inhibition of succinate dehydrogenase with a single dose of malonate, given at the onset of reperfusion, has long-term protective effects in mice subjected to transient coronary occlusion. Thus, malonate-treated mice showed a reduced area of scarring after myocardial infarction, an effect associated with significant decreases in interstitial and perivascular fibrosis in distant areas, improvements in left ventricular function, as assessed by echocardiography, and attenuated enhancements in expression of the fibrotic markers NF-κB and Smad2/3 in the remote myocardium. 

Previous studies have demonstrated that reversible succinate dehydrogenase inhibition during reperfusion with malonate is cardioprotective against acute myocardial infarction, both in isolated mouse hearts subjected to global ischemia followed by 1 h of reperfusion [15] and in pigs subjected to transient coronary occlusion and 2 h of reflow [16]. These effects were associated with the accumulation of succinate in the reperfused myocardium of treated animals, leading to attenuated reverse electron transfer to complex I, reduced ROS production, and lower MPTP opening [13,14,15]. However, hypothetically, it would be possible that these protective actions of malonate disappear in the long term. For this reason, in this study, we have evaluated the long-term effects of succinate dehydrogenase inhibition on scar size and post-infarction left ventricular remodeling. Adverse left ventricular remodeling is one of the most important consequences of acute myocardial infarction, leading to heart failure and a poor quality of life for affected patients [18]. 

First, it was needed to confirm the efficacy of malonate against myocardial infarction in our present model. In this case, we administered the succinate dehydrogenase inhibitor during initial reperfusion through a jugular vein at a concentration of 10 mg/kg/min. Under these conditions, disodium malonate was able to reduce infarct size by about 47%, a similar reduction to that previously described in other animal models [15,16]. Moreover, when we analyzed the long-term effects of a single dose of malonate, we were able to demonstrate that transient inhibition of succinate dehydrogenase during initial reperfusion reduced the area of scarring after infarction, an effect associated with a significant improvement in myocardial function 28 days after infarction. Furthermore, an attenuated enhancement in expression of the pro-inflammatory and fibrotic markers NF-κB and Smad2/3 was observed in the remote myocardium.

In contrast to NF-κB and Smad2/3, no changes were detected in the expression of the fibroblast differentiation/activation marker FAPα, whereas a reduction in αSMA, as compared with sham-operated mice, was observed 28 days after infarction in hearts from both malonate-treated and untreated animals, with no significant differences between them. As collagen deposition probably occurs during the first days following infarction, these data do not preclude the occurrence of enhanced differentiation of fibroblasts in the untreated animals early after injury. Further, contributions from smooth muscle cells may also explain these last findings. On the other hand, an RT-PCR analysis detected increases in several markers of fibrosis 28 days after myocardial infarction, as compared with sham-operated animals, but with no differences between saline- and malonate-treated mice. 

Despite previous studies that have nicely demonstrated that chronic treatment with malonate for 14 days promotes cardiomyocyte proliferation and cardiac regeneration 4 weeks after permanent coronary occlusion in adult mice [17], it is unlikely that a single dose, as that used in the present study, results in such effects. Instead, and as previously suggested [15,16,19,20], it is plausible that malonate in this setting would be reducing cell death, which is known to occur during the first minutes of reperfusion [21].

The main determinant of adverse post-infarction left ventricular remodeling is the size of the necrotic area [18]. However, different authors have postulated that, in addition to necrosis, apoptosis may also play an important role in cell death after myocardial infarction [22,23,24]. If this were the case, it may cause a late enhancement of the final infarct size. Under these conditions, it is plausible that treatments able to acutely reduce infarct size fail to have long-term protective effects against scar size and left ventricular remodeling. In contrast, our present results, in which transient succinate dehydrogenase inhibition is able to reduce the area of scar and post-infarction left ventricular remodeling, do not support this possibility. In fact, they are in accordance with other studies suggesting that apoptosis does not play an important role in cell death after an episode of myocardial ischemia–reperfusion. In this sense, we have previously demonstrated that cell death after myocardial ischemia–reperfusion predominantly occurs in the form of necrosis, as denoted by extensive propidium iodide staining of isolated mouse hearts subjected to ischemia and 1 h of reperfusion, but not by apoptosis, as demonstrated by the absence of positive cells using the TUNEL technique [15]. Furthermore, some studies have suggested that the caspase 3-dependent apoptotic pathway would not be operative in adult cardiomyocytes [25]. Indeed, genetic deletion of caspases-3 and -7 has not been associated with changes in infarct size, determined 24 h after coronary occlusion, nor in the scar area, measured 28 days after reperfusion in mice subjected to transient coronary ligature [26].

An important limitation of this work is that experiments were conducted using a mouse model of myocardial infarction. Whether our findings can be replicated in more clinically relevant and complex models, such as the pig model of transient coronary occlusion, warrants further investigation. Furthermore, in our experimental setup, malonate was administered at the onset of reperfusion. However, it is foreseeable that some patients may experience some treatment delays. The efficacy of malonate when administered later during reperfusion remains unknown. Lastly, the formulation utilized in our study, disodium malonate, is able to permeate cardiomyocyte membranes through monocarboxylate transporter 1 (MCT1), a process facilitated by the low pH environment present in the area at risk [19]. The use of ester prodrugs of malonate or even acidic formulations could potentially offer additional benefits [20].

In conclusion, our present results, in combination with those obtained in other animal models [15,16], strengthen the possibility that succinate dehydrogenase inhibition with malonate at the onset of reperfusion becomes a new therapeutic approach in patients with STEMI.

## 4. Materials and Methods

The present study conforms to the NIH Guide for the Care and Use of Laboratory Animals (NIH publications No. 85-23, revised 1996) and was performed in accordance with European legislation (Directive 2010/63/UE). The study was approved by the Ethics Committee of our institution (CEEA 35.17).

### 4.1. Animals

Thirty-one adult male C57BL/6J mice (10–12 weeks, 30–35 g) were anesthetized with ketamine (50 mg/kg, IP) and sodium pentobarbital (40 mg/kg, IP) [27]. After intubation and ventilation (SAR-830/P Ventilator, CWE Inc., Ardmore, PA, USA), the animals were placed on an adjustable heating pad to maintain a core temperature of 36–37 °C. Their hearts were exposed through a lateral thoracotomy at the fourth intercostal space, and the left anterior descending coronary artery (LAD) was ligated using a 6/0 silk snare (Ethincon Endo-surgery, Cincinnati, OH, USA), located 1 mm distal to the left atrial appendage. All animals were subjected to 45 min of LAD occlusion followed by reperfusion. During the ischemic period, an external jugular vein was dissected free and cannulated using a polyethylene PE-10 catheter. This catheter was used to continuously administer a saline solution with or without malonate at a dose of 10 mg/kg/min, given at a flow rate of 120 μL/kg/min. The infusion began during the last 5 min of ischemia and was extended for the first 15 min of reperfusion. This dose is in the same order of magnitude as those previously shown to be effective against acute ischemia–reperfusion injury in vivo [13,19]. Successful performance of LAD occlusion was verified visually by the appearance of a pallor tone in the distal myocardium and by elevation of the ST segment at the electrocardiogram. After reperfusion, the occluding snare was kept in place to allow later determination of the ligature position. Following surgery, all animals were treated with buprenorphine (0.05 mg/kg every 6 h, subcutaneously) during the first 48 h.

### 4.2. Experimental Protocols

First, to confirm that succinate dehydrogenase inhibition with malonate during reperfusion was also protective against myocardial infarction in our present model, 10 mice were subjected to LAD occlusion followed by 24 h of reperfusion (n = 5/group). Thereafter, in the remaining animals, the reperfusion phase was extended for 28 days to assess the effects of malonate on myocardial scars and post-infarction left ventricular remodeling.

### 4.3. Infarct Size and Myocardial Scar Measurements

Animals used to assess the acute effects of succinate dehydrogenase inhibition on myocardial infarction were sacrificed by a sodium pentobarbital overdose (100 mg/kg, IP). LAD was reoccluded, and the hearts were quickly removed and retrogradely perfused through the aorta with a Krebs solution in a Langendorff system (3 mL/min), as previously described [15]. Then, 0.5 mL of a 5% Evans Blue saline solution were given through a side port to delineate the area at risk. Immediately, hearts were removed from the system and cut into slices that were incubated in a 1% 2,3,5-triphenyltetrazolium chloride solution (37 °C, 15 min) to stain infarct size, as previously described (Figure 1A) [15]. Area at risk was expressed as a percentage of total ventricular weight, whereas infarct size was expressed as a percentage of area at risk.

The histological extent of the myocardial scar after 28 days of reperfusion was calculated as the percentage of the fibrotic area to the left ventricular area, as previously described [28]. At the end of the experimental protocol, animals were sacrificed by a sodium pentobarbital overdose (100 mg/kg, IP). Hearts were quickly excised, and after removal of both atria and great vessels, ventricles were weighted and divided into two parts. The basal half was immediately snap-frozen in liquid N_2_, whereas the apical area, just below the occluding snare, was fixed overnight with 4% paraformaldehyde, embedded in paraffin, and cut into 4 μm sections. Histological sections were then stained with Picrosirius Red (Sigma-Aldrich, Saint Louis, MO, USA) and scanned and evaluated using Image ProPlus 4.5 software (Media Cybernetics, Rockville, MD, USA).

### 4.4. Transthoracic Echocardiography

Echocardiographic measurements were performed in animals reperfused for 28 days, both at baseline and at the end of the experimental protocol, with a Vivid q portable ultrasound system using an ILS 12 MHz transducer (GE Healthcare, Chicago, IL, USA) applied to the shaved chest wall of mice lightly anesthetized with isoflurane (1–1.5%). Ejection fraction (EF), left ventricular end-diastolic internal diameter (LVEDD), left ventricular end-systolic internal diameter (LVESD), interventricular septum thickness (IVS), and posterior wall thickness (LVPW) were measured in M-mode recordings. Fractional shortening (FS) was calculated as (LVEDD-LVESD)/LVEDD × 100.

### 4.5. Western Blot Analysis

Myocardial samples from the basal half of the heart, located above the site of the coronary ligature, were snap-frozen and homogenized (Diax 600 homogenizer, Heidolph, Schwabach, Germany) in homogenization ice-cold RIPA buffer (NaCl 150 mmol/L, Trizma Base 50 mmol/L, Triton X-100 1%, SDS 0.1%, Tween20 0.1% (pH 8.0), containing phosphatase inhibitors (sodium fluoride 1 mmol/L and sodium orthovanadate 1 mmol/L), and a protease cocktail inhibitor (1%)). Myocardial samples from four additional sham-operated animals were also included. Protein lysates were obtained from the supernatant after centrifugation at 3600× *g* for 10 min (4 °C). Protein extracts were then electrophoretically separated on 10% polyacrylamide gels. Expression of markers involved in the pro-inflammatory and fibrotic responses, including NF-κB p65 (#ab16502, abcam, 1:1000), Smad2/3 (#sc-398844, Santa Cruz Biotechnology, 1:500), pSMAD2/3 (Thr8) (#TA325852, OriGene, 1:1000), transforming growth factor β1 (TGFβ1) (#ab92486, abcam, 1:1000), connective tissue growth factor (CTGF) (#ab6992, abcam, 1:1000), fibroblast activation factor α (FAPα) (#PIPA5120990, Invitrogen, 1:1000), and α-smooth muscle actin (αSMA) (#A5228, Sigma, 1:2000), was analyzed in remote myocardium using Western blot according to standard procedures [29,30]. Glyceraldehyde-3-phosphate dehydrogenase (GAPDH) (#GTX627408, GeneTex, 1:1000) was used as a loading control.

### 4.6. qPCR Analyses

Total RNA was extracted from myocardial samples obtained from the basal half of the heart, above the site of coronary ligature, using the Nucleospin RNA extraction kit (Macherey-Nagel, Düren, Germany). RNA was retrotranscribed into cDNA with a High-Capacity cDNA Reverse Transcription Kit (Applied Biosystems, Waltham, MA, USA). Gene expression was measured in duplicate with a 7900HT Fast Real-Time PCR System (Applied Biosystems, Waltham, MA, USA) using TaqMan Universal PCR master mix (Thermofisher, Waltham, MA, USA) and pre-designed gene-specific probes (Thermofisher, Waltham, MA, USA). Relative mRNA levels were determined using the comparative threshold method and expressed as fold change with respect to sham-operated animals. RPL19 was used as an endogenous control housekeeping gene.

### 4.7. Statistics

Data are expressed as mean ± SEM. Differences were assessed using a Student’s *t* test. ANOVA and Tukey tests were used to assess differences in expression of fibrotic markers by Western blot. A repeated measures ANOVA (MANOVA) was used to assess changes in echocardiographic or body weight data. Differences were considered significant at *p* < 0.05.

## 5. Conclusions

Reversible inhibition of succinate dehydrogenase with malonate at the onset of reperfusion has long-term protective effects in mice subjected to transient coronary occlusion. These effects are associated with an attenuated enhancement of pro-inflammatory and fibrotic markers in the remote myocardium. 

## Figures and Tables

**Figure 1 ijms-25-04366-f001:**
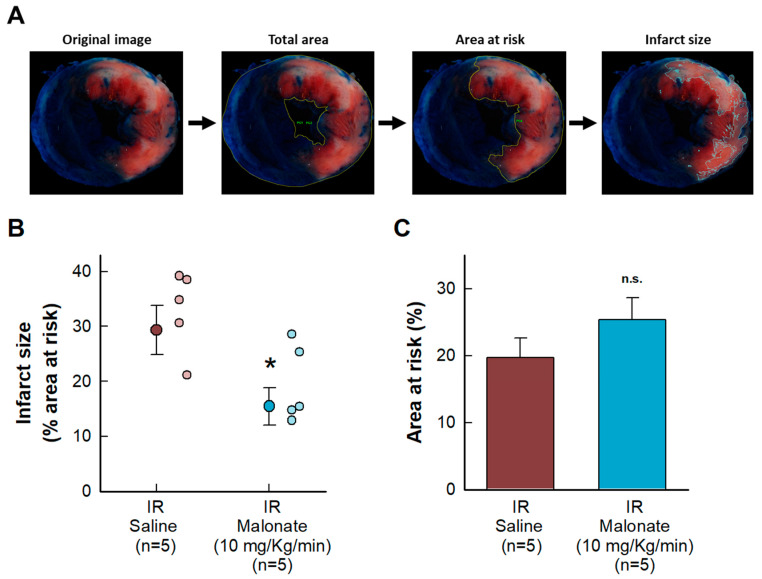
Effects of intravenous disodium malonate (10 mg/kg/min), given during the last 5 min of coronary occlusion and the first 15 min of reperfusion, on infarct size in mice subjected to 45 min of LAD occlusion and 24 h of reperfusion. (**A**) Method used to delineate the total slice area and the size of the area at risk and infarction. (**B**) Mean infarct size and individual values in both groups of animals. (**C**) Size of the area at risk. n.s. indicates no significant differences, whereas * (*p* < 0.05) indicates significant differences vs. control hearts (Student’s *t* test, n = 5/group).

**Figure 2 ijms-25-04366-f002:**
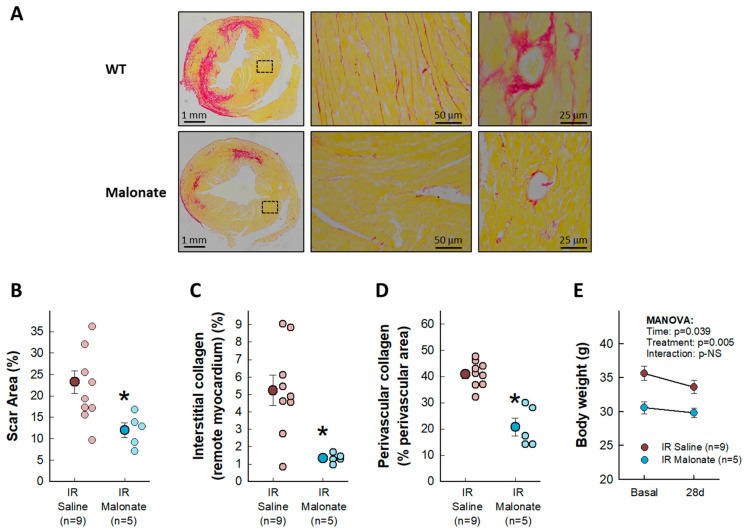
Effects of intravenous disodium malonate (10 mg/kg/min) given during the last 5 min of coronary occlusion and the first 15 min of reperfusion on scar area and interstitial and perivascular fibrosis in mice subjected to 45 min of LAD occlusion and 28 days of reperfusion. The upper panel (**A**) shows representative images of post-infarction scarring and interstitial and perivascular fibrosis in the remote myocardium, as assessed using Picrosirius Red staining, for a control mouse and a malonate-treated animal. Lower graphs depict mean and individual values for scar area (**B**), interstitial collagen deposition (**C**), and perivascular fibrosis (**D**) in the same groups of mice. * (*p* < 0.05) indicates significant differences vs. control hearts (Student’s *t* test, n = 9 for saline-treated animals and 5 for malonate-treated animals). (**E**) Changes in body weight in control animals and in malonate-treated mice. Repeated measures ANOVA demonstrated no interaction between time and treatment (n = 9 for saline-treated animals and 5 for malonate-treated animals).

**Figure 3 ijms-25-04366-f003:**
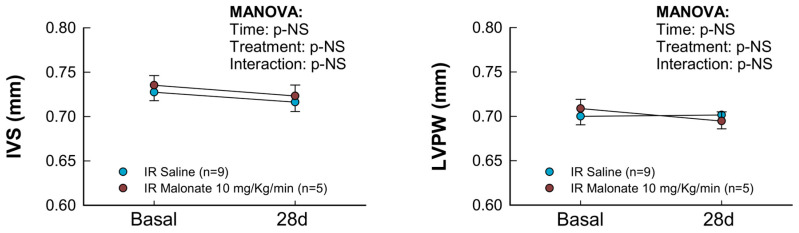
Changes in interventricular septum thickness (IVS), left ventricular posterior wall thickness (LVPW), left ventricular end-diastolic internal diameter (LVEDD), left ventricular end-systolic internal diameter (LVESD), ejection fraction (EF), and fractional shortening (FS) in mice subjected to 45 min of LAD occlusion and 28 days of reperfusion and treated with or without intravenous disodium malonate (n = 9 for saline-treated mice and 5 for malonate-treated animals). Statistical analysis was performed by repeated measures ANOVA that demonstrated a significant interaction between time and treatment for EF and FS, which indicates attenuated deterioration of myocardial function in malonate-treated mice.

**Figure 4 ijms-25-04366-f004:**
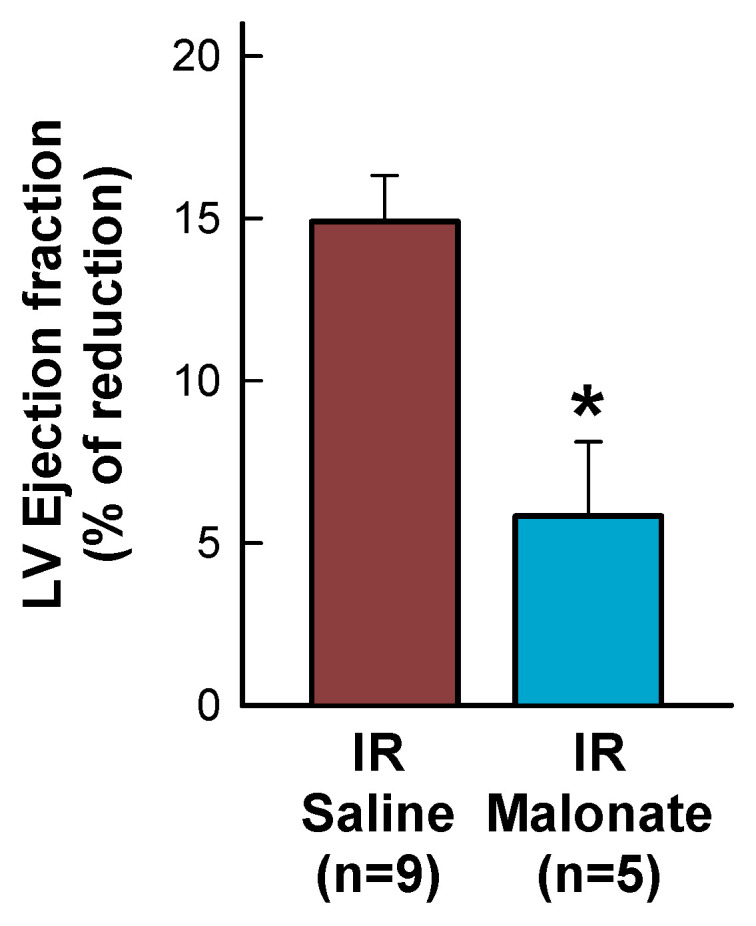
Percentage reduction in left ventricular (LV) ejection fraction, expressed with respect to baseline values, in mice subjected to 45 min of LAD occlusion and 28 days of reperfusion and treated with or without intravenous disodium malonate. * (*p* < 0.05) indicates significant differences between both groups (Student’s *t* test, n = 9 for saline-treated mice and 5 for malonate-treated animals).

**Figure 5 ijms-25-04366-f005:**
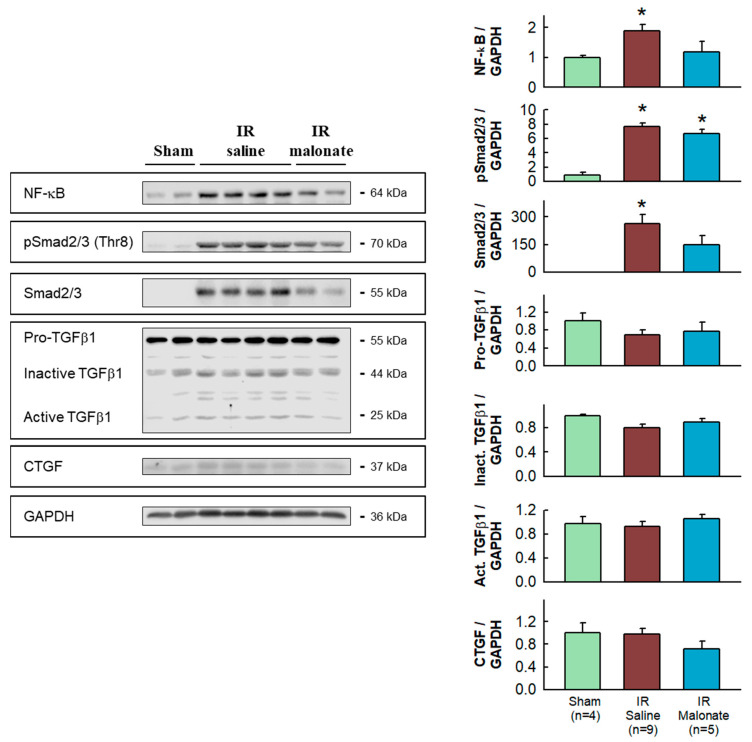
Western blot analysis of fibrotic markers in remote myocardium. Left panels show representative Western blots for NF-κB, Smad2/3 and pSmad2/3, TGFβ1, CGTF, and GAPDH in tissue extracts from sham-operated mice and from animals subjected to 45 min of LAD occlusion followed by 28 days of reperfusion and treated with or without malonate during initial reflow. The right panels show corresponding quantifications. * (*p* < 0.05) indicates significant differences vs. sham-operated mice (ANOVA and Tukey post-hoc tests, n = 4 for sham animals, 9 for saline-treated mice, and 5 for malonate-treated animals).

**Figure 6 ijms-25-04366-f006:**
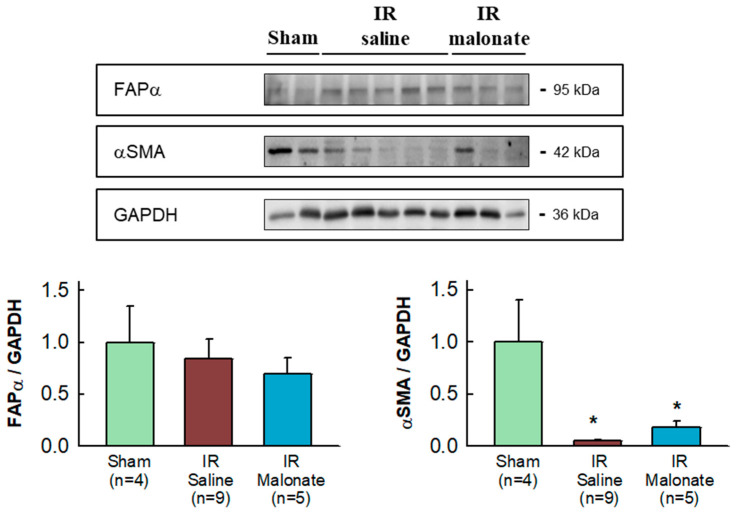
Western blot analysis of fibroblast differentiation markers in the remote myocardium. The upper image shows a representative Western blot for fibroblast activation factor α (FAPα) and α-smooth muscle actin (αSMA), together with GAPDH, in tissue extracts from sham-operated mice and from animals subjected to 45 min of LAD occlusion followed by 28 days of reperfusion and treated with or without malonate during initial reflow. The lower panels show corresponding quantifications. * (*p* < 0.05) indicates significant differences vs. sham-operated mice (ANOVA and Tukey post-hoc tests; n = 4 for sham animals, 9 for saline-treated mice, and 5 for malonate-treated animals).

**Figure 7 ijms-25-04366-f007:**
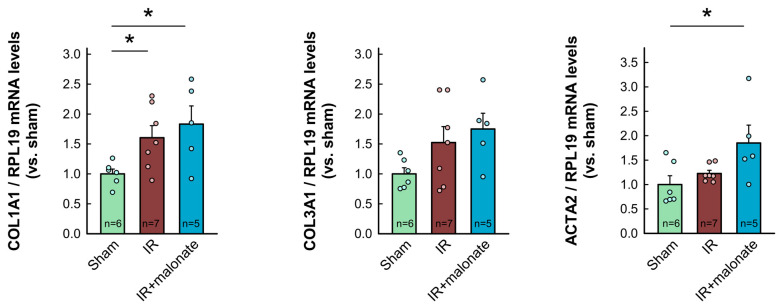
Myocardial levels of mRNAs coding for proteins involved in collagen synthesis (COL1A1, COL3A1, ACTA2, and TGFβ1) and in the inflammatory response (IL-1β and TNFα) in tissue extracts from sham-operated mice and from animals subjected to 45 min of LAD occlusion followed by 28 days of reperfusion and treated with or without malonate during initial reflow. Values are expressed as fold change with respect to RPL19, used as a housekeeping gene, and with respect to sham-operated animals. * (*p* < 0.05) indicates significant differences vs. indicated groups (ANOVA and Tukey post-hoc tests; n = 5–6 for sham animals, 7 for saline-treated mice, and 5 for malonate-treated animals).

## Data Availability

Data are contained within the article and available from the corresponding author upon reasonable request.

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
