# Peer review of "Long-Term Protective Effects of Succinate Dehydrogenase Inhibition during Reperfusion with Malonate on Post-Infarction Left Ventricular Scar and Remodeling in Mice"

_ijms, 2024, doi:10.3390/ijms25084366_

Round 1

Reviewer 1 Report

Comments and Suggestions for Authors

This article investigates the long-term protective effects of disodium malonate-mediated succinate dehydrogenase inhibition on myocardial infarction (MI) and post-infarction left ventricular remodeling in a murine model of transient coronary occlusion. The study explores the long term outcomes, examining infarct size, ventricular function, and molecular markers associated with inflammation and fibrosis. The results suggest that malonate treatment during reperfusion exerts sustained protective effects, highlighting its potential as a therapeutic strategy for patients with ST-segment elevation myocardial infarction (STEMI).

I have a few questions/suggestions (in order of magnitude)

·        In Figure 5, specify the statistical method used in the figure legend, as you have done for other figures

·        Add representative histology image for Sirius red analysis (Figure 2)

·        What is the percentage of interstitial and perivascular fibrosis in remote myocardium?

·        For Figure 5, authors should consider performing western blot for alpha-SMA with either periostin or FAPα (known marker for activated /differentiated fibroblast), to see if the drug reduced or promoted (de)differentiation of fibroblasts which resulted in fibrosis and scar size

·        At line 112, a trend towards an observed reduced enhancement in LVEDD and LVESD.. mention actual p value.

Author Response

We appreciate the reviewer for providing these constructive comments and suggestions. Below are our responses to the specific comments:

  • In Figure 5, specify the statistical method used in the figure legend, as you have done for other figures

Statistical tests used are now specified in each figure legend, including figure 5.

  • Add representative histology image for Sirius red analysis (Figure 2).

A representative histology image for Sirius red analysis has been added (new Figure 2), as suggested by the reviewer.

  • What is the percentage of interstitial and perivascular fibrosis in remote myocardium?

We agree with the reviewer on the importance of interstitial and perivascular fibrosis for post-infarct cardiac function. Consequently, we have included representative images of both types of fibrosis (new figure 2A), along with their quantification in Figures 2C and D. An explanation is given in lines 124-126 of the revised version of the manuscript.

  • For Figure 5, authors should consider performing western blot for alpha-SMA with either periostin or FAPα (known marker for activated /differentiated fibroblast), to see if the drug reduced or promoted (de)differentiation of fibroblasts which resulted in fibrosis and scar size.

According to reviewer’s suggestion we have conducted a new western blot analysis for both alpha-SMA and FAPalpha in the remote myocardium. The results are presented in lines 196-200 and in the new figure 6, and their interpretation is discussed in lines 269-276 of the new version of the manuscript. In addition, new figure 7 includes data obtained by qPCR (discussed in lines 276-278).

  • At line 112, a trend towards an observed reduced enhancement in LVEDD and LVESD.. mention actual p value.

We have decided to remove this sentence from the revised version of the manuscript, as p values were far from significance (0.622 and 0.278, respectively).

We hope these revisions address the reviewer's concerns satisfactorily. Thank you for your valuable feedback.

Reviewer 2 Report

Comments and Suggestions for Authors

This study by Laura Valls-Lacalle, presented data on the protective effect of Malonate (Succinate dehydrogenase inhibitor) in mouse model of myocardial infarction (I/R). Authors performed 45min LAD, followed by 24h (and 28day) reperfusion. Mice were perfused with malonate 15min of reperfusion stage. The authors performed different stainings to analyze the infarct area and scar formation. Additionally, cardiac function was assessed by echocardiography. Finally, authors found that malonate perfusion significantly reduced the infarct size, scar formation which were associated with increased ejection fraction (EF).  

Below are my concerns:

Please improve the introduction.

Figure 1: Its hard to understand figures. How did authors analyze infarct size and why did it was mentioned as (% of area at risk).

Figure 5: which part of heart tissue was used for WB analysis? I appreciate authors performing echocardiography; however, I must agree that it itself cannot support their hypothesis, particularly when the manuscript is submitted as an original research paper.

The data presented in this manuscript is poor compared with the similar papers published in the field (https://doi.org/10.1093/cvr/cvv279, https://doi.org/10.1161/CIRCULATIONAHA.120.049952). The authors have chosen limited methodological approaches, and the data presented here is too little for the original research paper.

In my opinion, the data presented in this manuscript may not be sufficient to fully support their hypothesis. Therefore, I recommend authors perform additional analysis and improve the quality of the presentation.

For example, since cardiac tissue is readily available, authors could perform IF stainings for macrophage phenotypes, IL-10, PDGFb, etc markers that are  associated with cardiac regeneration. Thereby can easily improve the manuscript. 

I regret to inform you that my decision is not in support of this manuscript in this present form.

Please carefully correct multiple typos before you resubmit elsewhere

Author Response

We thank the reviewer for his/her constructive comments and suggestions. Below are our responses to the specific comments:

Please improve the introduction.

We thank the reviewer for this suggestion. Accordingly, we have improved the introduction, including new information of importance in the field (lines 81-87), which is also discussed in lines 279-284.

Figure 1: Its hard to understand figures. How did authors analyze infarct size and why did it was mentioned as (% of area at risk).

As stated in the original version of the manuscript, infarct size was determined by TTC staining 24 h after coronary occlusion. You can find the complete protocol in the methods section (lines 346-353). In summary, following sacrifice by anesthetic overdose, the LAD was reoccluded, and hearts were quickly removed and retrogradely perfused through the aorta with a Krebs solution in a Langendorff system (3 mL/min). Then, 0.5 mL of a 5% Evans Blue saline solution were given through a side port to delineate the area at risk. Immediately, hearts were removed from the system and cut in slices, that were incubated in 1% 2,3,5-triphenyltetrazolium chloride solution (37ºC, 15 min), to stain infarct size, as previously described.

This method allows for the quantification of both the area at risk and infarct size, which is typically expressed as a percentage of the area at risk. Such methodology is widely accepted in the field for assessing infarct size shortly after infarction (please, refer to the article by Liepinsh et al., J Pharmacol and Toxicol Methods 2013;67:98-106, doi: 10.1016/j.vascn.2012.11.001).

To help the reader to understand how we measured infarct size, we have added, in Figure 1A, an schematic representation of the methodology used, as can be seen below:

Evans Blue staining of infarct

This method is different to that used several weeks after myocardial infarction. In the last case, myocardial infarction is determined by Picrosirius Red staining, measured by planimetry and expressed as percentage of left ventricular area (method used in figure 2).

Figure 5: which part of heart tissue was used for WB analysis? I appreciate authors performing echocardiography; however, I must agree that it itself cannot support their hypothesis, particularly when the manuscript is submitted as an original research paper.

Western blot analysis was performed in samples obtained from distant myocardium, located above the coronary ligature, in the basal half of the heart. This is now specified in the methods section, point 4.5.

While our echocardiographic data alone may not fully support our hypothesis, when viewed in conjunction with the reduction in scar size and interstitial fibrosis observed in malonate-treated animals, we believe that they lend support to the efficacy of this strategy in both reducing infarct size and mitigating adverse left ventricular remodeling.

The data presented in this manuscript is poor compared with the similar papers published in the field (https://doi.org/10.1093/cvr/cvv279, https://doi.org/10.1161/CIRCULATIONAHA.120.049952). The authors have chosen limited methodological approaches, and the data presented here is too little for the original research paper.

We appreciate this comment. In fact, the mechanisms of malonate protection after ischemia-reperfusion using a single dose have been widely explored, and our initial intention was to test whether this protective effect extends beyond the first hours of reperfusion.

Our situation is different to that in the paper by Bae et al (doi: doi.org/10.1161/CIRCULATIONAHA.120.049952). In their seminal paper, they nicely demonstrated that chronic treatment with malonate in an adult mice model of permanent coronary ligature promotes cardiomyocyte proliferation and cardiac regeneration. Given the importance these data have in the context of malonate cardioprotection, they are now explained in the introduction (lines 80-86) and discussed in lines 279-284. We apologize for the omission of this important paper. Anyway, in our present study we just gave a single dose of malonate to confirm or discard whether the initial protection against acute injury extends in the long-term. Under these conditions we believe that is unlikely that a single dose of malonate would promote cardiomyocyte proliferation. The use of a single dose is now emphasized throughout the manuscript.

Furthermore, we have added new mechanistic insights in the revised version of the manuscript. By Western Blot we have analyzed the expression of a-smooth muscle actin and FAPa, to assess whether our treatment modulates fibroblast activation/differentiation (new figure 6). In addition, by qPCR we have analyzed the expression of different markers of fibrosis and inflammation (new figure 7).

In my opinion, the data presented in this manuscript may not be sufficient to fully support their hypothesis. Therefore, I recommend authors perform additional analysis and improve the quality of the presentation.

For example, since cardiac tissue is readily available, authors could perform IF stainings for macrophage phenotypes, IL-10, PDGFb, etc markers that are associated with cardiac regeneration.

Thereby can easily improve the manuscript.

Thanks to the reviewer for this comment. We have performed new experiments to improve the quality of the manuscript. As commented above, we have analyzed by Western Blot the expression of markers of fibroblast activation/differentiation (new figure 6). In addition, by RT-PCR we have analyzed the expression of different markers of fibrosis and inflammation (new figure 7).

I regret to inform you that my decision is not in support of this manuscript in this present form.

We hope these revisions address the reviewer's concerns satisfactorily. Thank you for your valuable feedback.

Please carefully correct multiple typos before you resubmit elsewhere.

We have corrected different typos through the manuscript.

Round 2

Reviewer 2 Report

Comments and Suggestions for Authors

I appreciate the authors for revising the manuscript. There are few minor corrections I may suggest:

1) Fig1: If authors state no difference in 1c, the bar must indicate. 

2) The figure legends are poorly written. The statistical test whether t test or other must be indicated in all figures.  Please indicate n number. 

3) Please add a few key points of the study limitation at the end of the discussion. 

This manuscript can be accepted after the required corrections been made.

Author Response

I appreciate the authors for revising the manuscript. There are few minor corrections I may suggest:

We thank the reviewer for his/her constructive comments and suggestions. Below are our responses to the specific comments:

1) Fig1: If authors state no difference in 1c, the bar must indicate. 

Figure 1c now includes a statement (“n.s.”) indicating lack of significance, and it is explained in the figure legend.

2) The figure legends are poorly written. The statistical test whether t test or other must be indicated in all figures.  Please indicate n number. 

All figure legends have been checked and where possible we have clarified them. All statistical tests were already specified in the figure legends in the previous version. Regarding the n numbers, they appear now both in the figure and in the figure legend.

3) Please add a few key points of the study limitation at the end of the discussion.

A paragraph stating some limitations of our work now appears at the end of the discussion (lines 309-319).

We hope these revisions address the reviewer's concerns satisfactorily. Thank you for your valuable feedback.